# Defining explicit definitions of potentially inappropriate prescriptions for antidiabetic drugs in patients with type 2 diabetes: A systematic review

Erwin Gerard[1,2]*, Paul Quindroit[1], Madleen Lemaitre[3,4], Laurine Robert[1,2], Sophie Gautier[5], Bertrand Decaudin[2,6], Anne Vambergue[3,7], Jean-Baptiste Beuscart[1]

**1** Univ. Lille, CHU Lille, ULR 2694 - METRICS: Évaluation des Technologies de Santé et des Pratiques Médicales, Lille, France, **2** CHU Lille, Institut de Pharmacie, Lille, France, **3** CHU Lille, Department of Diabetology, Endocrinology, Metabolism and Nutrition, Lille University Hospital, Lille, France, **4** University of Lille, Lille, France, **5** CHU de Lille, Centre Régional de Pharmacovigilance, Lille, France, **6** Univ. Lille, CHU Lille, ULR 7365 - GRITA: Groupe de Recherche sur les Formes Injectables et les Technologies Associées, Lille, France, **7** European Genomic Institute for Diabetes, University School of Medicine, Lille, France

* erwin.gerard.etu@univ-lille.fr

**Data Availability Statement:** All relevant data are within the paper and its Supporting information files.

## Abstract

### Introduction

Potentially inappropriate prescriptions (PIPs) of antidiabetic drugs (ADs) (PIPADs) to patients with type 2 diabetes mellitus (T2DM) have been reported in some studies. The detection of PIPs in electronic databases requires the development of explicit definitions. This approach is widely used in geriatrics but has not been extended to PIPADs in diabetes mellitus. The objective of the present literature review was to identify all explicit definitions of PIPADs in patients with T2DM.

### Materials and methods

We performed a systematic review of the literature listed on Medline (*via* PubMed), Scopus, Web of Science, and, Embase between 2010 and 2021. The query included a combination of three concepts ("T2DM" AND "PIPs" AND "ADs") and featured a total of 86 keywords. Two independent reviewers selected publications, extracted explicit definitions of PIPADs, and then classified the definitions by therapeutic class and organ class.

### Results

Of the 4,093 screened publications, 39 were included. In all, 171 mentions of PIPADs (corresponding to 56 unique explicit definitions) were identified. More than 50% of the definitions were related to either metformin (34%) or sulfonylureas (29%). More than 75% of the definitions were related to either abnormal renal function (56%) or age (22%). In addition, 20% (n = 35) mentions stated that biguanides were inappropriate in patients with renal dysfunction and 17.5% (n = 30) stated that sulfonylureas were inappropriate above a certain age. The definitions of PIPADs were heterogeneous and had various degrees of precision.

**Funding:** This research was funded by PreciDIAB, which is jointly supported by the French National Agency for Research (ANR- 18- IBHU- 0001), by the European Union (FEDER - agreement NP0025517), by the Hauts- de- France Regional Council (agreement 20001891/NP0025517) and by the European Metropolis of Lille (MEL, agreement 2019_ESR_11).

**Competing interests:** The authors have declared that no competing interests exist.

**Abbreviations:** AD, antidiabetic drugs; ADE, adverse drug event; ALP, alkaline phosphatase; ALT, alanine aminotransferase; ASP, aspartate aminotransferase; ATC, anatomical therapeutic chemical; CKD-EPI, chronic kidney disease epidemiology collaboration; DPP-4, dipeptidyl peptidase-4; eGFR, estimated glomerular filtration rate; GGT, gamma-glutamyl transferase; GLP-1 RA, glucagon-like peptide-1 receptor agonist; ICD-9-CM, international classification of diseases, ninth revision, clinical modification; KDIGO, kidney disease improving global outcomes; MDRD, modification of diet in renal disease; MeSH, medical subject heading; PIP, potential inappropriate prescription; PIPAD, potential inappropriate prescription related to antidiabetic drugs; PRISMA, preferred reporting items for systematic reviews and meta-analyses; SGLT-2, sodium-glucose transport protein 2; STOPP, screening tool of older persons' prescriptions; T2DM, type 2 diabetes mellitus.

## Conclusion

Our results showed that researchers focused primarily on the at-risk situations related to biguanide prescriptions in patients with renal dysfunction and the prescription of sulfonylureas to older people. Our systematic review of the literature revealed a lack of consensus on explicit definitions of PIPADs, which were heterogeneous and limited (in most cases) to a small number of drugs and clinical situations.

## 1 Introduction

For patients with type 2 diabetes mellitus (T2DM), most antidiabetic drugs (ADs) other than insulin (*i.e.* drugs in the A10B class, according to the Anatomical Therapeutic Chemical Classification System(ATC)) are prescribed by general practitioners or other primary care physicians [1]. There are clear guidelines on the prescription of ADs to patients with T2DM. However, according to a study conducted in all the community hospitals in 43 US states, the likelihood of an adverse drug event (ADE) in a patient taking an AD was 9.7% and was associated with an annual cost of $2.59 billion [2]. In 2002, the estimated cost per patient per year in Europe was €2834; hence, the appropriate use of ADs might reduce the costs associated with the long-term complication [3]. Hence, potential inappropriate prescriptions (PIPs) of ADs to patients with T2DM appear to constitute a major issue.

The likelihood of PIPs can be reduced by applying two very different types of approach. Implicit approach is based on an expert judgment of the quality of care with regard to the patient's condition and the medical literature [4,5]. In contrast, explicit approach does not require a direct expert assessment and is based on the prescription data used and implemented for use in medical informatics [4,5]. The assessment of the appropriateness of ADs prescriptions is generally based on an implicit approach, with reference to guidelines (*e.g.* those issued by the American Diabetes Association or the European Association for the Study of Diabetes) or medical practice [6,7]. Moreover, a mixed implicit-explicit approach has recently been recommended as a way of minimizing PIPs [8,9].

In explicit approach, the detection of PIPs is usually based on definitions. Hence, a PIP is defined as drug use with a poor risk-benefit ratio–particularly when safer alternatives exist–and can be classified as underuse, overuse, or misuse [10–13]. PIPs are associated with elevated morbidity and mortality and have major financial consequences by requiring hospital admission or prolonging hospital stays, partly due to the occurrence of ADEs [2,14]. Most of the lists of PIPs in the literature have been developed for older people, using validated criteria such as the Beers criteria [15] and the Screening Tool of Older Persons' Prescriptions (STOPP) criteria [16]. However, there are only five definitions of PIPs related to ADs (PIPADs) in older people. Although PIPADs in patients with T2DM have been reported in a number of studies [17–22], no validated criteria for these prescriptions have yet been published.

The objective of this systematic review of the literature was to list explicit definitions of PIPADs and thus improve the prescription of ADs by physicians who are not diabetologists.

## 2 Materials and methods

### 2.1 Search strategy

A method for developing explicit definitions of PIPADs in patients with T2DM has been published previously [23]. The present systematic review was reported in accordance with the

Preferred Reporting Items for Systematic Reviews and Meta-Analyses (PRISMA) guidelines [24,25]. The protocol was registered in the PROSPERO database (reference: CRD42021250028). Methodological assistance was provided by three librarians at the University of Lille (Lille, France). At each stage, the results were validated by four reviewers (A.V., M. L., J-B.B., and, S.G.). We systematically searched publications in the Medline (*via* PubMed), Embase, Web of Science and, Scopus databases. Studies were limited to those published in English and French between January 2010 and June 2021. Publications before 2010 were not included because drug strategies for the management of diabetes have changed particularly rapidly over the last ten years. Furthermore, 2010 corresponds to the year in which new classes of AD (such as glucagon-like peptide-1 receptor agonists (GLP-1 RAs) and dipeptidyl peptidase-4 (DPP-4) inhibitors) were widely prescribed by primary care physicians and specialist physicians.

Our search strategy covered three topics: (i) diabetes and diabetic patients, (ii) PIPs, (iii) ADs. The search terms included two Medical Subject Heading (MeSH) terms or keywords for diabetes and diabetics patients, 21 for PIPs, and 63 ADs (S1 Table). The combination of these search terms generated a total of 2,646 queries.

This systematic review did not include a meta-analysis or a risk of bias assessment; the review's objective was solely to list published explicit definitions, regardless of the type of study.

## 2.2 Study selection

The Medline (*via* PubMed), Scopus, Web of Science and, Embase search results (containing each publication's title, author(s), journal, and, digital object identifier) were exported to an Excel file (Microsoft Corporation, Redmond, WA, USA). The publication titles were initially screened independently by two reviewers (E.G. and P.Q.), and 10% of the sample were selected at random and checked by a third reviewer (M.L.). Firstly, records were screened on the basis of the title. The exclusion criteria were as follows: studies in types of diabetes other than T2DM (*e.g.* type 1 diabetes, gestational diabetes, and secondary diabetes), studies in non-human models (*e.g.* cells and mice), and studies of medications other than ADs (*e.g.* insulins).

Secondly, two reviewers (E.G. and P.Q.) assessed publications for possible retrieval by analyzing the abstract. The exclusion criteria were as follows: studies in types of diabetes other than T2DM (*e.g.* type 1 diabetes and gestational diabetes), studies in non-human models (*e.g.* cells and mice), and studies of medications other than ADs (*e.g.* insulins).

Lastly, publications were assessed for eligibility. The aim was to identify publications in which there was at least one mention of an explicit definition of a PIPAD in patients with T2DM. The exclusion criteria were as follows: drug class not specified, studies involving patients with T2DM but not ADs, or the absence of an explicit definition. A definition was considered to be implicit (*i.e.* not explicit) if it was based on (i) guidelines or expert knowledge or (ii) a dosage adjustment based on a laboratory parameter but for which quantitative criteria were lacking (*e.g.* adjustment of the dose to the patient's level of renal function but without any mention of the creatinine or estimated glomerular filtration rate (eGFR) cut-off. A definition was considered to be explicit if it was based on prescription data and did not require expert knowledge. For example, the STOPP criteria explicitly state that glibenclamide is inappropriate for adults 65 years of age or older [16].

## 2.3 Data extraction

Full manuscripts were obtained for all titles and abstracts that met the inclusion criteria and were then coded with NVivo software (version 12, QSR International Pty Ltd. Australia 2020).

Two reviewers (E.G. and P.Q.) independently examined the full text and extracted any mentions of explicit definitions. A third reviewer (M.L.) was called upon to resolve any differences of opinion and helped to form a consensus. At this stage, the following records were excluded: studies that did not include T2DM, studies of animals or cultured cells, pharmacological studies, studies about mechanisms of action, and definitions that considered patients with T2DM but did not consider ADs.

## 2.4 Classification and aggregation of PIPADs mentions into definitions

The mentions of explicit definitions were then classified into groups by two independent reviewers (E.G. and P.Q.), according to the drug (in the ATC Classification System) and the organ (heart, liver, kidney, *etc.*). For example, all mention of definitions for metformin and renal failure that were deemed to be similar were grouped together. For example: "*metformin is inappropriate with an eGFR < 30 ml/min/1.73 m²*" and "*metformin is inappropriate with an eGFR < 30 ml/min calculated with the calculated with the chronic kidney disease epidemiology collaboration (CKD-EPI) equation*" were aggregated into "*metformin is inappropriate with an eGFR < 30 ml/min*". A third reviewer (M.L.) was called upon to resolve any differences of opinion. Similar mentions were then aggregated into definitions and validated by four reviewers (A.V., M.L., J-B.B., and, S.G.).

## 3 Results

### 3.1 Selection of studies

The set of 2,646 queries generated 4,093 non-duplicate hits. The selection process identified 39 publications with at least one mention of an explicit definition of PIPADs and that met the criteria for inclusion in the systematic review. The PRISMA flowchart is shown in Fig 1. The full search strategy is given in S1 Table.

### 3.2 Classification of mentions of PIPADs

A total of 171 mentions of explicit definitions of PIPADs were extracted from the 39 publications. The full list of word-for-word mentions is given in S2 Table. The mentions were classified into six domains related respectively to age, renal dysfunction, heart failure, liver dysfunction, pancreas dysfunction, and, other conditions (Table 1). The "other conditions" included lung dysfunction, diabetic ketoacidosis, lactic acidosis, dehydration, drug-drug interactions, hypoglycemia, cognitive impairment, at-risk occupations, and, obesity.

Two clinical situations related to PIPADs were very frequently mentioned by researchers: 94 (56%) of the mentions were related to renal dysfunction, and 38 (22%) were related to an age limit. Consequently, more than 75% (n = 132) of the mentions defined PIPADs with regard to age and renal dysfunction. Most of the mentions related to age (n = 33 out of 38) were in publications on PIPs in elderly people (*i.e.* the STOPP criteria [16], the Beers criteria [15], and the "PRescribing Optimally in Middle-aged People's Treatments" criteria [26]). Two drugs were very frequently mentioned in definitions of PIPADs: 57 (34%) concerned metformin and 40 (29%) concerned sulfonylureas. These two clinical situations and two drugs were combined in many definitions of PIPADs. For example, 20% (n = 35) mentions stated that biguanides were inappropriate in patients with renal dysfunction and 17.5% (n = 30) stated that sulfonylureas were inappropriate above a certain age.

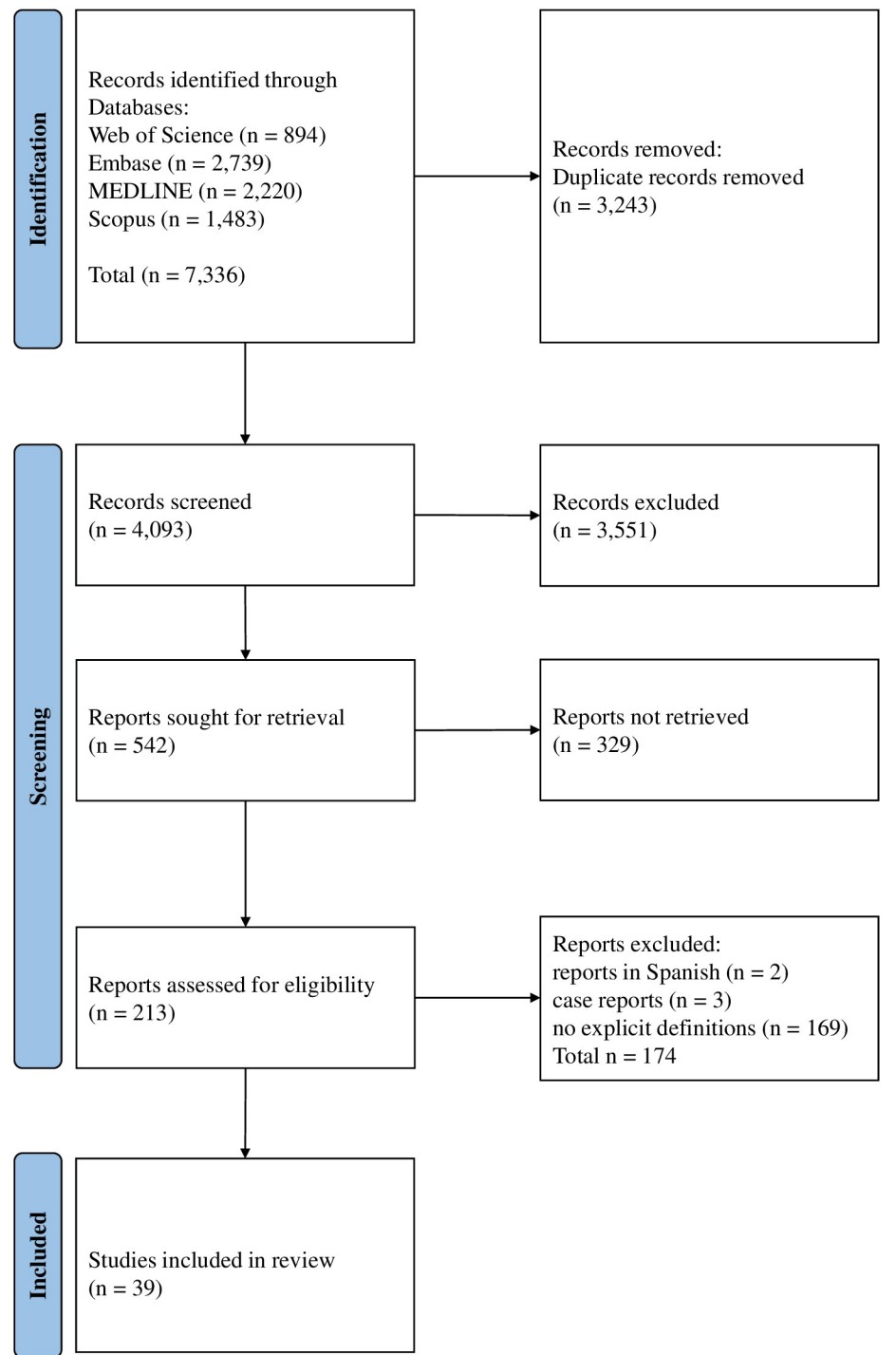

**Fig 1. PRISMA flow diagram, describing the selection and screening process.**

### 3.3 Aggregation of mentions into definitions

Similar mentions of PIPADs were aggregated into unique definitions. All the mentions of PIPADs related to renal function used an eGFR threshold. Most researchers used the

**Table 1. The number of explicit mentions of PIPADs, classified by organ or by patient age.**

| Drug class | Renal dysfunction | Age | Heart failure | Liver dysfunction | Pancreas dysfunction | Other conditions | Total |
|---|---|---|---|---|---|---|---|
| Biguanides | n = 35 | n = 3 | n = 4 | n = 6 | n = 1 | n = 8 | 57 |
| Thiazolidinediones | - | n = 2 | n = 5 | n = 1 | - | n = 1 | 9 |
| Glinides | n = 1 | - | - | - | - | - | 1 |
| Sulfonylureas | n = 12 | n = 30 | n = 1 | n = 1 | - | n = 5 | 49 |
| Glucagon-like peptide-1 receptor agonists | n = 12 | n = 1 | - | - | n = 2 | - | 15 |
| Dipeptidyl peptidase-4 inhibitors | n = 11 | - | - | - | n = 3 | n = 1 | 15 |
| Alpha glucosidase inhibitors | n = 3 | - | - | - | - | - | 3 |
| Sodium-glucose transport protein 2 inhibitors | n = 13 | n = 1 | - | - | - | - | 14 |
| Associations of antidiabetic drugs | n = 7 | - | - | - | - | - | 7 |
| Oral diabetic agents (except metformin) | - | n = 1 | - | - | - | - | 1 |
| Total | 94 | 38 | 10 | 8 | 6 | 15 | 171 |

For example, 35 mentions stated that biguanides are potentially inappropriate in patients with renal dysfunction.

n = number of mentions.

thresholds of the Kidney Disease Improving Global Outcomes (KDIGO). The 94 mentions of PIPADs related to renal function were therefore grouped into 24 explicit definitions of PIPADs. The results are summarized in Table 2, in which each green cell corresponds to an explicit definition of PIPADs. It appeared that a broad range of thresholds were suggested. For instance, seven different thresholds of eGFR were suggested for PIPs of biguanides; this resulted in seven different definitions of PIPADs.

The mentions of PIPADs related to age used five different age thresholds, from an age under 18 years to an age over 85. The 38 age-related mentions of PIPADs were therefore grouped into 11 explicit definitions. Results are summarized in Table 3, in which each green cell corresponds to an explicit definition of PIPADs. It appears that most of the publications (n = 27) suggested a threshold of 65 years and over for sulfonylureas (based on the STOPP

**Table 2. Explicit definitions of PIPADs by renal failure stage, according to the KDIGO definition and classification system for acute kidney injury.**

| Drug class | eGFR < 15 ml/min | eGFR < 30 ml/min | eGFR < 45 ml/min | eGFR < 60 ml/min | eGFR < 90 ml/min | eGFR ≥ 90 ml/min | Other conditions | Total |
|---|---|---|---|---|---|---|---|---|
| Biguanides | n = 3 | n = 15 | n = 3 | n = 9 | n = 2 | n = 1 | n = 2 | 35 |
| Glinides | - | - | - | - | - | - | n = 1 | 1 |
| Sulfonylureas | - | n = 7 | n = 3 | - | - | - | n = 2 | 12 |
| Glucagon-like peptide-1 receptor agonists | n = 1 | n = 10 | n = 1 | - | - | - | - | 12 |
| Dipeptidyl peptidase-4 inhibitors | n = 2 | n = 3 | n = 5 | n = 1 | - | - | - | 11 |
| Alpha glucosidase inhibitors | - | n = 3 | - | - | - | - | - | 3 |
| Sodium-glucose transport protein 2 inhibitors | - | - | n = 6 | n = 6 | - | - | n = 1 | 13 |
| Associations of antidiabetic drugs | - | - | - | n = 1 | - | - | n = 6 | 7 |
| Total | 6 | 38 | 18 | 17 | 2 | 1 | 12 | 94 |

Each green cell corresponds to an explicit definition of PIPADs. For example, "the prescription of biguanides are potentially inappropriate when the eGFR is below 15 mL/min".

n = number of mentions; eGFR = estimated glomerular filtration rate.

**Table 3. Explicit definitions of PIPADs by age group.**

| Drug class | Age < 18 | Age > 45 | Age > 60 | Age > 65 | Age > 75 | Age > 80 | Age > 85 | Total |
|---|---|---|---|---|---|---|---|---|
| Biguanides | - | - | - | - | - | n = 2 | n = 1 | 3 |
| Thiazolidinediones | n = 1 | - | - | - | n = 1 | - | - | 2 |
| Sulfonylureas | | n = 1 | n = 1 | n = 27 | n = 1 | - | - | 30 |
| Glucagon-like peptide-1 receptor agonists | - | - | - | - | n = 1 | - | - | 1 |
| Sodium-glucose transport protein 2 inhibitors | - | - | - | - | n = 1 | - | - | 1 |
| Oral diabetic agents (except metformin) | - | - | - | - | - | n = 1 | - | 1 |
| Total | 1 | 1 | 1 | 27 | 4 | 3 | 1 | 38 |

Each green cell corresponds to an explicit definition of PIPADs. For example "Thiazolidinediones are potentially inappropriate in patients over the age of 75".

n = number of mentions for each explicit definition.

criteria [16]), although one publication suggested a threshold of 75 and over. Two age thresholds were also suggested for biguanides.

The 39 mentions not related to renal function or age were aggregated into 21 unique definitions (S3 Table). After aggregation and validation by the steering committee, the 171 mentions gave rise to a total of 56 unique definitions.

## 3.4 Heterogeneity of mentions related to the same definition of PIPADs

The mentions related to each definition are detailed in S2 Table. It can be seen that the written formulation of a mention related to a given definition of PIPADs varied from one publication to another. For the definition of biguanides, for the example, researchers estimated renal insufficiency with several different methods (creatine clearance with the Cockcroft-Gault formula, or the eGFR according to the Modification of Diet in Renal Disease (MDRD) equation or the CKD-EPI equation) or did not specify the method.

For a given definition, the level of precision varied from one mention to another. For example, 5 mentions were identified for the definition "Metformin is inappropriate in patients with liver dysfunction". One mention was simply "*liver dysfunction*", whereas another defined liver dysfunction as "*an elevation of the liver enzyme activity ((alanine aminotransferase (ALT), aspartate aminotransferase (ASP)) >3-fold the normal range and, the presence of common symptoms of liver dysfunction*", and another stated "*Metformin is inappropriate in patients with severe hepatic dysfunction, defined as biochemical evidence of hypoalbuminemia and abnormal serum levels of at least two of the following: total bilirubin, alanine aminotransferase (ALT), alkaline phosphatase (ALP) and*, gamma-glutamyl transferase (GGT)".

Lastly, 22% of the mentions (n = 39: 17 for renal dysfunction, 8 for age, and 14 for other conditions) leading to a definition were found only once, i.e. in a single study.

## 4 Discussion

We reviewed the literature on explicit definitions of PIPADs. Based on mentions from 39 sources, we recorded a total of 56 explicit definitions. Our results showed that researchers focused primarily on the at-risk situations related to biguanides prescriptions in patients with renal dysfunction and the prescription of sulfonylureas to older people. The eGFR and/or age thresholds in explicit definitions and the definition's level of precision and written formulation were very heterogeneous. Furthermore, many explicit definitions were mentioned in a single publication only. Our results therefore revealed a lack of consensus on explicit definitions of PIPADs.

Many studies have highlighted the risks associated with prescribing metformin to patients with renal failure, including the risk of lactic acidosis [27–29]. As expected, we found a large number of definitions of PIPADs related to this situation [30–32]. Likewise, the risk of ADEs with sulfonylureas in older people (with a high risk of hypoglycemia and secondary complications) has been widely reported [33–37] and has been widely publicized through the STOPP criteria [16,38]. Our results showed that the most likely at-risk situations have been identified and suggested as PIPs by many researchers. In contrast, other classes of AD were less well represented in the literature definitions of PIPs: DPP-4 inhibitors (n = 15, including 11 related to renal dysfunction), GLP-1 RAs (n = 15, including 12 related to renal dysfunction), and sodium-glucose transport protein 2 (SGLT-2) inhibitors (n = 14 including 13 related to renal dysfunction). However, GLP-1 RAs are associated with hypoglycemic ADEs, and SGLT-2 inhibitors and DPP-4 inhibitors are associated with cardiovascular ADEs [39–43]. Furthermore, most of the clinical situations associated with a risk of ADEs were related to age and renal insufficiency only; we did not identify any explicit definitions related to hypoglycemia or glycated hemoglobin levels.

Our results highlighted the lack of consensus and the heterogeneity of the definitions. In the literature, there was no agreement on the eGFR cut-off that influenced the prescription of metformin. In fact, we noted seven different eGFR thresholds at which metformin was potentially inappropriate. Ten publications suggested that GLP-1 RAs are inappropriate when the eGFR is < 30 ml/min (n = 10 mentions per definition), whereas another publication suggested an eGFR < 15 ml/min (n = 1) and yet another suggested an eGFR < 45 ml/min (n = 1). This variety might be due to the publications' geographical diversity and different publication dates. Furthermore, the eGFR thresholds were not standardized, and several estimators were used (*e.g.* the CKD-EPI equation and the Cockcroft-Gault formula) [44]. Concerning the age criteria, most of explicit definitions were based on the Beers criteria or the STOPP criteria (*i.e.* in older patient populations) [15,16]. However, some publications suggested new age thresholds that had not been validated by an expert group. Furthermore, for a given definition of PIPADs, the level of precision, written formulation and, domain varied from one mention to another. Co-morbidities were described with a variable degree of precision. For example: "*Thiazolidinediones are contraindicated in patients with a history of heart failure*" was less precise than "*Thiazolidinediones are inappropriate in patients with a history of hospitalization for heart failure (~1 hospitalization with a main diagnosis of heart failure (International Classification of Diseases, Ninth Revision, Clinical Modification (ICD-9-CM) code = 402.01, 402.11, 402.91, 404.01, 404.11, 404.91, or 428)) before the thiazolidinedione was prescribed*". The definition was sometimes vague; for example, "*Metformin is inappropriate in patients with respiratory dysfunction*" specified neither the type of respiratory dysfunction nor whether the dysfunction was acute or chronic. Lastly, many explicit definitions of PIPADs were quoted only once, in a single publication. Some of these definitions of PIPADs were surprising, such as "*oral diabetic agents (except metformin) are potentially inappropriate in patients aged 80 and over*" [45] (Table 3) and "*Metformin is not recommended for use after the age of 80*" [46] (Table 3). These definitions of PIPADs are not consistent with current guidelines and should probably not be promoted. Definitions of PIPADs quoted only one should therefore be considered with caution and should be validated by expert consensus.

## The value of explicit definitions

Our present results highlight the need to establish an expert consensus on PIPAD definitions that can be translated explicitly (via a qualitative study and a Delphi survey) and applied to patients with T2DM. Next to the systematic review, the qualitative study will aim to identify as

many explicit definitions as possible, then the Delphi survey will aim to provide a consensus among them as show as in our published protocol [23]. Most strategies for prescribing ADs are based on guidelines and expert's opinion with implicit definitions. Worldwide, many rational, evidence-based, consensual guidelines have been published. The diffusion of validated explicit PIPADs could assist in the prescribing of ADs for non-diabetologist physicians. This approach has never been used in diabetology whereas it has been successful in other domains, such as in the geriatrics, to improve the appropriateness and fight against ADEs in older people with complex drug regimens [47–49].

## Limitations

Firstly, the search was limited to the Medline *via* PubMed, Embase, Web of Science and, Scopus database. A second limitation is the possible omission of certain keywords and MeSH terms, some PIPADs definitions might not therefore have been found. Thirdly, the review period started on January 2010 and ended in June 2021, i.e. more than six months before submission of the manuscript. However, we checked that an update did have not an impact on the present results. Fourthly, the most frequently found definitions of PIPADs were based on older patients or patients with renal insufficiency, whereas T2DM is not limited to these patient profiles. Fifthly, we focused solely on drug therapy in patients with T2DM, regarding insulin therapy, it is less easy to define explicitly the risks of inappropriate prescriptions. Indeed, their prescriptions are regularly subject to expert advice, their dosage is variable both inter-individually (e.g. weight, age) and intra-individually (e.g. time of day, physical activity). Lastly, the selected studies came from various countries, in which the definitions of PIPADs sometimes depend on guidelines issued by national authorities and learned societies. For example, thiazolidinediones are no longer authorized in France but are available in the United States. The overall management of patients with T2DM will require specific, in-depth work in the future.

## 5 Conclusion

Our systematic review identified 56 explicit definitions of PIPADs (excluding insulins) in patients with T2DM. The scope of the definitions was often limited to age, renal function, biguanides, and sulfonylureas. Many definitions of PIPADs were suggested only once in a single publication, and the definitions were generally very heterogeneous.

A list of explicit definitions of PIPADs based on expert consensus is needed to improve the prescribing of ADs by physicians who are not diabetologists. The next steps of our work will therefore consist in completing this list of PIPADs in patients with T2DM with expert opinions by a qualitative study. Then, the Delphi survey will aim to provide a consensus among them.

## Supporting information

**S1 Table. Search strategies for Medline (via PubMed), Web of Science, Scopus, and Embase.**
(PDF)

**S2 Table. Mentions of explicit definitions of PIPADs extracted from the publications.**
(PDF)

**S3 Table. Aggregated definitions of PIPADs not related to renal function or age.**
(PDF)

## Acknowledgments

The authors thank Laurence Crohem, Anne-Sophie Guilbert and Julien Meignotte (University Library, University of Lille, Lille, France) for helping to develop the database queries.

The authors thank David Fraser (Biotech Communication) for English editing.

## Author Contributions

**Conceptualization:** Erwin Gerard, Paul Quindroit, Anne Vambergue, Jean-Baptiste Beuscart.

**Data curation:** Erwin Gerard, Paul Quindroit.

**Formal analysis:** Erwin Gerard.

**Funding acquisition:** Erwin Gerard, Paul Quindroit, Madleen Lemaitre.

**Methodology:** Erwin Gerard, Paul Quindroit, Anne Vambergue.

**Project administration:** Paul Quindroit, Jean-Baptiste Beuscart.

**Supervision:** Anne Vambergue, Jean-Baptiste Beuscart.

**Validation:** Laurine Robert, Sophie Gautier, Bertrand Decaudin, Anne Vambergue, Jean-Baptiste Beuscart.

**Visualization:** Jean-Baptiste Beuscart.

**Writing – original draft:** Erwin Gerard, Paul Quindroit.

**Writing – review & editing:** Madleen Lemaitre, Laurine Robert, Sophie Gautier, Bertrand Decaudin, Anne Vambergue, Jean-Baptiste Beuscart.

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
