## [Decision Letter · Decision Letter 0]

20 Jul 2022

PONE-D-22-09991Defining explicit definitions of potentially inappropriate prescriptions for antidiabetic drugs in patients with type 2 diabetes: a systematic reviewPLOS ONE

Dear Dr. GERARD,

Thank you for submitting your manuscript to PLOS ONE. After careful consideration, we feel that it has merit but does not fully meet PLOS ONE’s publication criteria as it currently stands. Therefore, we invite you to submit a revised version of the manuscript that addresses the points raised during the review process.

We look forward to receiving your revised manuscript.

Kind regards,

Sairah Hafeez Kamran, PhD

Academic Editor

PLOS ONE

Journal Requirements:

Reviewers' comments:

Reviewer's Responses to Questions

**Comments to the Author**

1. Is the manuscript technically sound, and do the data support the conclusions?

Reviewer #1: Partly

Reviewer #2: Yes

Reviewer #3: Yes

2. Has the statistical analysis been performed appropriately and rigorously? 

Reviewer #1: Yes

Reviewer #2: Yes

Reviewer #3: Yes

3. Have the authors made all data underlying the findings in their manuscript fully available?

Reviewer #1: Yes

Reviewer #2: Yes

Reviewer #3: Yes

4. Is the manuscript presented in an intelligible fashion and written in standard English?

Reviewer #1: No

Reviewer #2: No

Reviewer #3: Yes

5. Review Comments to the Author

Reviewer #1: Thank you for submitting your manuscript. It is a strong manuscript, but it has minor mistakes.

1. It has grammatical and spacing issues that can be modified with an English language evaluation service.

2. There are a few spelling mistakes in this manuscript.

For example: in line#209 (hypoalbuminemia) and line#213 Table-1 (other conditions and antidiabetic).

3.. A lot of punctuation errors in line# 99, 113, 160, 202, 208, 224, 235, 262, 286, 288, and 477.

4. Table 1 has not been mentioned in the result section of the main manuscript of the article.

5. The following things are not justifiable: The abstract result section does not mention which antidiabetic drug is more related to renal dysfunction. Similarly, which one antidiabetic drug has contraindicated from which older-age diabetic patients. It has stated in the rest of the article but not in the abstract.

6. please insert the reference of criteria and guidelines in the whole article.

7. The conclusion of a manuscript is not focused. It is not clear what would be the next step in the context of this work.

The references should be uniform. Page numbers are missing about Pg#13.

Reviewer #2: In the present study, the authors have done a systematic review on inappropriate prescriptions for antidiabetic drugs in patients with type 2 diabetes. The work is well written and easy to follow. I would recommend the manuscript can be accepted in its present form.

Reviewer #3: This review article "Potentially inappropriate prescriptions (PIPs) of antidiabetic drugs (ADs) (PIPADs) to

40 patients with type 2 diabetes mellitus (T2DM)" based on the reports in the studies published in the period between the period of 2010 to 2021 using various scientific search engines. The manuscript is well-written and methodology is very sound. However , there a few important points which need to be addressed before its final acceptance to increase the readers interest.

1. In discussion section, the authors may include recommendations for future research perspective or highlight the grey area and practices based on findings of this reviewed literature.

2. All abbreviations should be expand (written in full) on the first appearance and then use appropriately throughout the manuscript.

3. Sentences should not be used with the abbreviations.

4. There are few typographical errors as well. Authors must go thorough the whole manuscript and rectify all the mistakes. A few examples are given below.

Line 133: No space before reference number [16]

Line 157, 163: (Error! Reference source not found.).

References 4, 7, 20, 35, and 40 are not according to journal format.

Tables 4,5 and 6 does not appear in the manuscript , whereas has been confused with figures 1, 2 and 3 in the manuscript.

Why sentence "Error! Reference source not found " in line 158 and 163 is bold?

6. PLOS authors have the option to publish the peer review history of their article (what does this mean?). If published, this will include your full peer review and any attached files.

Reviewer #1: **Yes: **Mehwish Mushtaq, Ph.D. Scholar, University of Peshawar; CRA in Metrics Research Organization, Pakistan

Reviewer #2: **Yes: **Nikhl Agrawal

Reviewer #3: **Yes: **Dr. Sheryar Afzal

---

## [Author Response · Author response to Decision Letter 0]

23 Aug 2022

Response to Reviewer 1 Comments

We thank Reviewer 1 for comments and constructive suggestions. We have carefully revised the manuscript according to each comment and suggestion.

Point 1. It has grammatical and spacing issues that can be modified with an English language evaluation service.

There are indeed spacing issues with double spaces and spaces before the ; or :. 

We thank the reviewer for his careful proofreading and we have used a software to make corrections of double spacing, spaces before “;” and “:”.

In figure 1, we corrected one spacing issue after ‘records removed’.

In the supplementary table S2, we corrected 5 spacing before “:” However, the 5 spacing errors came from 5 different quotes from our selected articles line#40, line#85, #line100, line#109 and, #153.

In the supplementary table S3, we corrected one spacing before “:”.#line26.

For grammar, we used the services of Biotech Communication (https://www.biotechcommunication.com/en/home/) for English Editing and Mr. David Fraser is mentioned in the acknowledgements. (line#331).

The manuscript has been corrected by a native English speaker (D.Phil. in Biochemistry from the University of Oxford), who has copy-edited more than 1000 manuscripts over the last 15 years. Most often, any remaining language issues are minor questions of personal preference and style (rather than grammar, idiom or clarity). Mr. Fraser is available to review grammatical formulations that seems incorrect for the reviewer. We would however need the exact sentences to be corrected. 

Point 2. There are a few spelling mistakes in this manuscript.

For example: in line#209 (hypoalbuminemia) and line#213 Table-1 (other conditions and antidiabetic).

We apologize for the errors, and we thank the reviewer for this comment. We have corrected these spelling mistakes in the revised version of the manuscript.

Point 3. A lot of punctuation errors in line# 99, 113, 160, 202, 208, 224, 235, 262, 286, 288, and 477.

We thank you for your attention during your review. We corrected the errors mentioned in the new version of the manuscript.

Point 4. Table 1 has not been mentioned in the result section of the main manuscript of the article.

Table 1 was correctly mentioned in the result section in the Word version of the manuscript. However, the conversion of the Word manuscript into PDF broke the link between table 1 and the reference.

We thank the reviewer for identifying this mistake and we apologize that we didn’t see it while checking the PDF during the submission. We corrected these errors in the new version of the manuscript.

Point 5. The following things are not justifiable: The abstract result section does not mention which antidiabetic drug is more related to renal dysfunction. Similarly, which one antidiabetic drug has contraindicated from which older-age diabetic patients. It has stated in the rest of the article but not in the abstract.

Indeed, our results showed many mentions of biguanide prescriptions in patients with renal dysfunction (20%) and prescriptions of sulfonylureas to older people (17.5%). This point is detailed in the manuscript results but is not specified in the abstract, as the reviewer points out.

We agree that mentioning biguanides are more related to renal dysfunction and sulfonylureas are more related to older-age diabetic patients would increase the relevance of our abstract.

In the revised version of the manuscript, we have provided the following information:

• Abstract, page 3, line #59 “In addition, 20% (n = 35) mentions stated that biguanides were inappropriate in patients with renal dysfunction and 17.5% (n = 30) stated that sulfonylureas were inappropriate above a certain age”

• And line #63 “Our results showed that researchers focused primarily on the at-risk situations related to biguanide prescriptions in patients with renal dysfunction and the prescription of sulfonylureas to older people.”

Point 6. please insert the reference of criteria and guidelines in the whole article.

We have now inserted the reference of criteria and guidelines in the whole document so, we added references in lines#178, #245, #264.

Point 7. The conclusion of a manuscript is not focused. It is not clear what would be the next step in the context of this work.

We agree with the reviewer that the conclusion of our manuscript could be more focused. We already planned the next steps of our work, and even published the protocol and several steps[1]. Our present results highlight the need to establish an expert consensus on PIPAD definitions. The systematic review and the qualitative study will aim to identify as many explicit definitions as possible, then the Delphi survey will aim to provide a consensus among them. 

The national project PreciDIAB (https://www.precidiab.org/en/), in collaboration with national and international partners, aims to determine a list of PIPs and to evaluate the impact of using these rules in clinical decision support systems in a pragmatic trial over 5 years.

In the revised version of the manuscript, we have now provided the following information:

• Discussion, Page 14, line #284: “Next to the systematic review, the qualitative study will aim to identify as many explicit definitions as possible, then the Delphi survey will aim to provide a consensus among them as show as in our published protocol”.

• And, Conclusion, page 16, line# 313: “The next steps of our work will therefore consist in completing this list of PIPADs in patients with T2DM with expert opinions by a qualitative study. Then, the Delphi survey will aim to provide a consensus among them”.

1. Quindroit P, Baclet N, Gerard E, Robert L, Lemaitre M, Gautier S, et al. Defining Potentially Inappropriate Prescriptions for Hypoglycaemic Agents to Improve Computerised Decision Support: A Study Protocol. Healthcare (Basel) 2021;9(11):1539. 

The references should be uniform. Page numbers are missing about Pg#13.

We apologize for the errors, we fixed them in the new version of manuscript. 

Response to Reviewer 2 Comments

Reviewer #2: In the present study, the authors have done a systematic review on inappropriate prescriptions for antidiabetic drugs in patients with type 2 diabetes. The work is well written and easy to follow. I would recommend the manuscript can be accepted in its present form.

We thank Reviewer 2 for comments. 

Response to Reviewer 3 Comments

We thank Reviewer 3 for comments and constructive suggestions. We have carefully revised the manuscript according to each comment and suggestion.

Point 1. In discussion section, the authors may include recommendations for future research perspective or highlight the grey area and practices based on findings of this reviewed literature.

Our present results highlight the need to establish an expert consensus on PIPAD definitions that can be translated explicitly and applied to patients with T2DM. The systematic review and the qualitative study will aim to identify as many explicit definitions as possible. Here, we intend to conduct a qualitative study to complete the definitions identified in the systematic review. Then, the Delphi survey will aim to gather opinions, build consensus among experts, and reduce the number of explicit definitions to a priority list.

This type of definition could be easily integrated into computerized decision support tools for the automated detection of PIPs and the re-evaluation by a clinical pharmacist. 

The national project PreciDIAB (https://www.precidiab.org/en/), in collaboration with national and international partners, aims to determine a list of PIPs and to evaluate the impact of using these rules in clinical decision support systems in a pragmatic trial over 5 years.

In the revised version of the manuscript, we have now provided the following information:

• Discussion, Page 14, line #284: “Next to the systematic review, the qualitative study will aim to identify as many explicit definitions as possible, then the Delphi survey will aim to provide a consensus among them as show as in our published protocol”

• And, Conclusion, page 16, line# 313: “The next steps of our work will therefore consist in completing this list of PIPADs in patients with T2DM with expert opinions by a qualitative study. Then, the Delphi survey will aim to provide a consensus among them”

Point 2. All abbreviations should be expand (written in full) on the first appearance and then use appropriately throughout the manuscript.

Point 3. Sentences should not be used with the abbreviations.

We thank the reviewer for this comment. We agree that #line156 we used abbreviation “CKD-EPI” whereas we defined this abbreviation line #206. Hence, we corrected and defined “CKD-EPI” line#156 instead of line#206. And, line#270 we did not defined ICD for International Classification of Diseases, we fixed it in the new version of manuscript. 

We corrected line#70 the use of abbreviation: “Anatomical Therapeutic Chemical Classification System (ATC)” instead of “ATC (Anatomical Therapeutic Chemical) classification system”. Then, we used abbreviation line#153 “ATC” instead of “Anatomical Therapeutic Chemical”

We already defined ADE as adverse drug event twice line #72 and line#90. We corrected for defined once #line73 and we corrected “ADE”, instead of “adverse drug event (ADEs)” line #90 in the new version of the manuscript.

In addition, we used in the main text, 21 abbreviations.

Among the most used abbreviation, “PIPAD” defined line #93 is correctly used more than 40 times. The abbreviations “AD”, “T2DM” and “PIP” are defined, respectively, lines #69, #69 and #77 are correctly used in the following manuscript, respectively 18 times, 16 times and 13 times.

The abbreviation “eGFR” is defined line#140 and used 13 times in the main text and several times in table 2.

Also, abbreviations “ATC” (defined line#70), “PRISMA” (defined line #103), “MeSH” (defined line#115), “KDIGO” (defined line#187) or ICD-9-CM (defined line#270) are used 3 times or less but there were commonly used.

At last, abbreviations “MDRD”, “ALT”, “ASP”, “ALP” or “GGT” (defined respectively, lines#206, #211, #212, #215, #216) are defined but not used in the following text because they come from quotations of articles resulting from our systematic review.

Il faut créer une légende avec uniquement les abréviations utilisées dans le supplementary data

Point 4. There are few typographical errors as well. Authors must go through the whole manuscript and rectify all the mistakes. A few examples are given below.

Line 133: No space before reference number [16]

We apologize for this. We fixed it in the new version of manuscript.

Line 157, 163: (Error! Reference source not found.).

Table 1 was correctly mentioned in the result section in the Word version of the manuscript. However, the conversion of the Word manuscript into PDF broke the link between table 1 and the reference.

References 4, 7, 20, 35, and 40 are not according to journal format.

We corrected the references according to the journal format. In lines #340, #349, #387, #431, #447 of the new version of manuscript.

Tables 4,5 and 6 does not appear in the manuscript, whereas has been confused with figures 1, 2 and 3 in the manuscript.

The conversion of the Word manuscript into PDF broke the link between tables and figures and their references. The numbers 4, 5, 6 were automatically created by the submission software. 

We thank the reviewer for identifying this mistake and we apologize that we didn’t see it. We corrected these errors in the new version of the manuscript.

---

## [Editor Report · Decision Letter 1]

25 Aug 2022

Defining explicit definitions of potentially inappropriate prescriptions for antidiabetic drugs in patients with type 2 diabetes: a systematic review

PONE-D-22-09991R1

Dear Dr. GERARD,

We’re pleased to inform you that your manuscript has been judged scientifically suitable for publication and will be formally accepted for publication once it meets all outstanding technical requirements.

Kind regards,

Sairah Hafeez Kamran, PhD

Academic Editor

PLOS ONE

---

## [Editor Report · Acceptance letter]

1 Sep 2022

PONE-D-22-09991R1 

Defining explicit definitions of potentially inappropriate prescriptions for antidiabetic drugs in patients with type 2 diabetes: a systematic review 

Dear Dr. Gerard:

I'm pleased to inform you that your manuscript has been deemed suitable for publication in PLOS ONE. Congratulations! Your manuscript is now with our production department. 

Kind regards, 

on behalf of

Dr. Sairah Hafeez Kamran 

Academic Editor

PLOS ONE